# ADVERSARIAL DOMAIN ADAPTATION FOR STABLE BRAIN-MACHINE INTERFACES

**Ali Farshchian, Juan A. Gallego, Lee E. Miller & Sara A. Solla**
Northwestern University, Evanston, IL, USA
{a-farshchiansadegh, juan.gallego, lm, solla}@northwestern.edu

**Joseph P. Cohen & Yoshua Bengio**
University of Montreal, Montreal, Canada
cohenjos@iro.umontreal.ca, yoshua.bengio@umontreal.ca

## ABSTRACT

Brain-Machine Interfaces (BMIs) have recently emerged as a clinically viable option to restore voluntary movements after paralysis. These devices are based on the ability to extract information about movement intent from neural signals recorded using multi-electrode arrays chronically implanted in the motor cortices of the brain. However, the inherent loss and turnover of recorded neurons requires repeated recalibrations of the interface, which can potentially alter the day-to-day user experience. The resulting need for continued user adaptation interferes with the natural, subconscious use of the BMI. Here, we introduce a new computational approach that decodes movement intent from a low-dimensional latent representation of the neural data. We implement various domain adaptation methods to stabilize the interface over significantly long times. This includes Canonical Correlation Analysis used to align the latent variables across days; this method requires prior point-to-point correspondence of the time series across domains. Alternatively, we match the empirical probability distributions of the latent variables across days through the minimization of their Kullback-Leibler divergence. These two methods provide a significant and comparable improvement in the performance of the interface. However, implementation of an Adversarial Domain Adaptation Network trained to match the empirical probability distribution of the residuals of the reconstructed neural signals outperforms the two methods based on latent variables, while requiring remarkably few data points to solve the domain adaptation problem.

## 1 INTRODUCTION

Individuals with tetraplegia due to spinal cord injury identify restoration of hand function as their highest priority (Anderson, 2004). Over 50% of respondents with a C1-C4 injury would be willing to undergo brain surgery to restore grasp (Blabe et al., 2015). Brain-Machine Interfaces (BMIs) aim to restore motor function by extracting movement intent from neural signals. Despite its great promise, current BMI technology has significant limitations. A BMI that maps neural activity in primary motor cortex (M1) onto motor intent commands should ideally provide a stable day-to-day user experience. However, the gradual alterations of the activity recorded by chronically implanted multi-electrode arrays, due to neuron turnover or electrode movement and failure (Barrese et al., 2013), causes considerable variation in the actions produced by the BMI. This turnover may occur within a single day (Downey et al., 2018), and is estimated to be on the order of 40% over two weeks (Dickey et al., 2009). In the face of changing neural signals, performance can be maintained by daily retraining the interface, but this is not a viable solution as it requires the user to keep on adapting to a new interface (Ajiboye et al., 2017).

There is a high degree of correlation across the M1 neural signals. This redundancy implies that the dimensionality of the underlying motor command is much lower than the number of M1 neurons, and even lower than the number of recorded M1 neurons (Gallego et al., 2017). The use of dimen-

sionality reduction methods is thus a common practice in BMI design, as it provides a more compact and denoised representation of neural activity, and a low-dimensional predictor of movement intent. Most of the earlier work used linear dimensionality reduction methods such as Principal Components Analysis (PCA) and Factor Analysis (FA) (Yu et al., 2009; Shenoy et al., 2013; Sadtler et al., 2014; Gallego et al., 2017); more recently, autoencoders (AEs) have been used for the nonlinear dimensionality reduction of neural signals (Pandarinath et al., 2018). Here we develop a deep learning architecture to extract a low-dimensional representation of recorded M1 activity constrained to include features related to movement intent. This is achieved by the simultaneous training of a deep, nonlinear autoencoder network based on neural signals from M1, and a network that predicts movement intent from the inferred low-dimensional signals. We show that this architecture significantly improves predictions over the standard sequential approach of first extracting a low-dimensional, latent representation of neural signals, followed by training a movement predictor based on the latent signals.

To stabilize the resulting BMI against continuous changes in the neural recordings, we introduce a novel approach based on the Generative Adversarial Network (GAN) architecture (Goodfellow et al., 2014). This new approach, the Adversarial Domain Adaptation Network (ADAN), focuses on the probability distribution function (PDF) of the residuals of the reconstructed neural signals to align the residual's PDF at a later day to the PDF of the first day the BMI was calculated. The alignment of residual PDFs results in the alignment of the PDFs of the neural data and of their latent representation across multiple days. We show that this method results in a significantly more stable performance of the BMI over time than the stability achieved using several other domain adaptation methods. The use of an ADAN thus results in a BMI that remains stable and consistent to the user over long periods of time. A successful domain adaptation of the neural data eliminates the need for frequent recalibration of the BMI, which remains fixed. This strategy is expected to alleviate the cognitive burden on the user, who would no longer need to learn novel strategies to compensate for a changing interface.

## 2  RELATED WORK

Current approaches to solving the stability problem for BMIs based on spiking activity recorded using chronically implanted multi-electrode arrays include gradually updating interface parameters using an exponentially weighted sliding average (Orsborn et al., 2012; Dangi et al., 2013), adjusting interface parameters by tracking recording nonstationarities (Zhang & Chase, 2013; Bishop et al., 2014) or by retrospectively inferring the user intention among a set of fixed targets (Jarosiewicz et al., 2015), cancelling out neural fluctuations by projecting the recorded activity onto a very low-dimensional space (Nuyujukian et al., 2014), training the interface with large data volumes collected over a period of several months to achieve robustness against future changes in neural recordings (Sussillo et al., 2016), and a semi-unsupervised approach based on aligning the PDF of newly predicted movements to a previously established PDF of typical movements (Dyer et al., 2017).

Other approaches, more similar to ours, are based on the assumption that the relationship between latent dynamics and movement intent will remain stable despite changes in the recorded neural signals. Recent studies reveal the potential of latent dynamics for BMI stability. Kao et al. (2017) use past information about population dynamics to partially stabilize a BMI even under severe loss of recorded neurons, by aligning the remaining neurons to previously learned dynamics. Pandarinath et al. (2018) extract a single latent space from concatenating neural recordings over five months, and show that a predictor of movement kinematics based on these latent signals is reliable across all the recorded sessions.

## 3  EXPERIMENTAL SETUP

A male rhesus monkey (*Macaca mulatta*) sat in a primate chair with the forearm restrained and its hand secured into a padded, custom fit box. A torque cell with six degrees of freedom was mounted onto the box. The monkey was trained to generate isometric torques that controlled a computer cursor displayed on a screen placed at eye-level, and performed a 2D center-out task in which the cursor moved from a central target to one of eight peripheral targets equally spaced along a circle centered on the central target (Figure 1A) (Oby et al., 2012).

To record neural activity, we implanted a 96-channel microelectrode array (Blackrock Microsystems, Salt Lake City, Utah) into the hand area of primary motor cortex (M1). Prior to implanting the array, we intraoperatively identified the hand area of M1 through sulcal landmarks, and by stimulating the surface of the cortex to elicit twitches of the wrist and hand muscles. We also implanted electrodes in 14 muscles of the forearm and hand, allowing us to record the electromyograms (EMGs) that quantify the level of activity in each of the muscles. Data was collected in five experimental sessions spanning 16 days. All methods were approved by Northwestern Universitys IACUC and carried out in accordance with the *Guide for the Care and Use of Laboratory Animals*.

## 4 METHODS

### 4.1 COMPUTATIONAL INTERFACE

Our goal is to reliably predict the actual patterns of muscle activity during task execution, based on the recorded neural signals. Similar real-time predictions of kinematics are the basis of BMIs used to provide control of a computer cursor or a robotic limb to a paralyzed person (Taylor et al., 2002; Hochberg et al., 2006; Collinger et al., 2013). Predictions of muscle activity have been used to control the intensity of electrical stimulation of muscles that are temporarily paralyzed by a pharmacological peripheral nerve block (Ethier et al., 2012), a procedure that effectively bypasses the spinal cord to restore voluntary control of the paralyzed muscles. Similar methods have been attempted recently in humans (Bouton et al., 2016; Ajiboye et al., 2017).

The BMI is a computational interface that transforms neural signals into command signals for movement control, in this case the EMG patterns. Here we propose an interface that consists of two components, a neural autoencoder (AE) and an EMG predictor (Figure 1B). The AE is a fully connected multilayer network consisting of an input layer, five layers of hidden units, and an output layer. The reconstruction aims at minimizing the mean square error (MSE) between input and output signals. Units in the latent and output layers implement linear readouts of the activity of the preceding layer. Units in the remaining hidden layers implement a linear readout followed by an exponential nonlinearity. To provide inputs to the AE, we start with neural data $s_t$ consisting of the spike trains recorded from $n$ electrodes at time $t$. We bin neural spikes at 50 ms intervals and apply a Gaussian filter with a standard deviation of 125 ms to the binned spike counts to obtain $n$-dimensional smoothed firing rates $x_t$. The output layer provides $n$-dimensional estimates $\hat{x}_t$ of the inputs $x_t$. The latent layer activity $z_t$ consist of $l$ latent variables, with $l < n$.

The EMG data $y_t$ is the envelope of the muscle activity recorded from $m$ muscles, with $m < n$. The $l$-dimensional latent activity $z_t$ is mapped onto the $m$-dimensional EMGs through a long-short term memory (LSTM) (Hochreiter & Schmidhuber, 1997) layer with $m$ units, followed by a linear layer $\hat{y}_t = W^T LSTM(z_t)$. To train the model, we minimize a loss function $\mathcal{L}$ that simultaneously accounts for two losses: $\mathcal{L}^x : \mathbb{R}^n \to \mathbb{R}^+$ is the MSE of the reconstruction of the smooth firing rates, and $\mathcal{L}^y : \mathbb{R}^m \to \mathbb{R}^+$ is the MSE of the EMG predictions:

$$\mathcal{L} = \lambda \mathcal{L}^x + \mathcal{L}^y = \frac{1}{T} \sum_{t=1}^{T} \left( \lambda ||\hat{x}_t - x_t||^2 + ||\hat{y}_t - y_t||^2 \right) \tag{1}$$

Here $T$ is the number of time samples, and $t$ labels the time-ordered data points that constitute the training set. The factor $\lambda$ that multiplies the AE loss adjusts for the different units and different value ranges of firing rates and EMGs; it equalizes the contributions of the two terms in the loss function so that the learning algorithm does not prioritize one over the other. The value of $\lambda$ is updated for each new training iteration; it is computed as the ratio $\lambda = \frac{\mathcal{L}^y}{\mathcal{L}^x}$ of the respective losses at the end of the preceding iteration. Once the neural AE and the EMG predictor networks have been trained on the data acquired on the first recording session, indicated as day-0, their weights remain fixed.

### 4.2 DOMAIN ADAPTATION

To stabilize a fixed BMI, we need to align the latent space of later days to that of the first day, when the fixed interface was initially built. This step is necessary to provide statistically stationary inputs to the EMG predictor. We first use two different approaches to align latent variables across days: Canonical Correlation Analysis (CCA) between latent trajectories and Kullback-Leibler (KL)

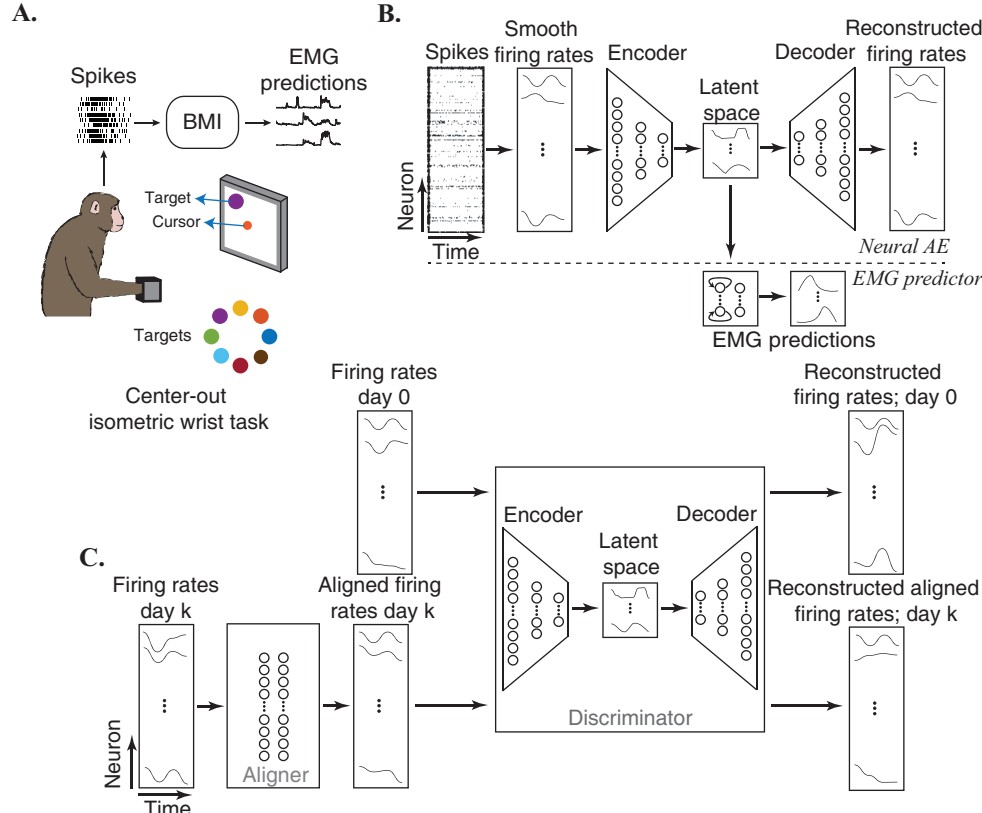

Figure 1: Experimental setup and methods. **A.** The isometric wrist center-out task with its eight targets, color coded. BMI schematics: recorded neural activity predicts muscle activity. **B.** The BMI consists of two networks: a neural AE and an EMG predictor. Recorded neural activity is binned and smoothed to provide an input to the AE. The activity of the low-dimensional latent space provides an input to the predictor of muscle activity. **C.** The ADAN architecture that aligns the firing rates of day-$k$ to those of day-0, when the BMI was built.

divergence minimization between latent distributions. We then use an ADAN to align the distributions of the residuals of the reconstructed neural data. This procedure results in the alignment of the distributions of the neural data and of their latent representation across days.

### 4.2.1 CANONICAL CORRELATION ANALYSIS (CCA)

Consider the latent activities $\boldsymbol{Z}_0$ corresponding to day-0 and $\boldsymbol{Z}_k$ corresponding to a later day-$k$; the AE is fixed after being trained with day-0 data. Both $\boldsymbol{Z}_0$ and $\boldsymbol{Z}_k$ are matrices of dimension $l$ by $8\tau$, where $l$ is the dimensionality of the latent space and $\tau$ is the average time duration of each trial; the factor of 8 arises from concatenating the averaged latent activities for each of the eight targets. The goal of CCA is to find a linear transformation of the latent variables $\boldsymbol{Z}_k$ so that they are maximally correlated with a linear transformation of the latent variables $\boldsymbol{Z}_0$ (Bach & Jordan, 2002). This well established approach, involving only linear algebra, has been successfully applied to the analysis of M1 neural data (Sussillo et al., 2015; Gallego et al., 2018; Russo et al., 2018). In summary, the analysis starts with a $QR$ decomposition of the transposed latent activity matrices, $\boldsymbol{Z}_0^T = \boldsymbol{Q}_0\boldsymbol{R}_0$, $\boldsymbol{Z}_k^T = \boldsymbol{Q}_k\boldsymbol{R}_k$. Next, we construct the inner product matrix of $\boldsymbol{Q}_0$ and $\boldsymbol{Q}_k$, and perform a singular value decomposition to obtain $\boldsymbol{Q}_0^T\boldsymbol{Q}_k = \boldsymbol{U}\boldsymbol{S}\boldsymbol{V}^T$. The new latent space directions along which correlations are maximized are given by $\boldsymbol{M}_0 = \boldsymbol{R}_0^{-1}\boldsymbol{U}$, and $\boldsymbol{M}_k = \boldsymbol{R}_k^{-1}\boldsymbol{V}$, respectively.

The implementation of CCA requires a one-to-one correspondence between data points in the two sets; this restricts its application to neural data that can be matched in time across days. Matching is

achieved here through the repeated execution of highly stereotypic movements; the correspondence is then established by pairing average trials to a given target across different days. In a real-life scenario, motor behaviors are not structured and moment-to-moment movement intent is less clear, interfering with the possibility of establishing such a correspondence. Alignment using CCA requires a supervised calibration through the repetition of stereotyped tasks, but ideally the alignment would be achieved based on data obtained during natural, voluntary movements. A successful unsupervised approach to the alignment problem is thus highly desirable.

### 4.2.2 KULLBACK-LEIBLER DIVERGENCE MINIMIZATION (KLDM)

For the unsupervised approach, we seek to match the probability distribution of the latent variables of day-$k$ to that of day-0, without a need for the point-to-point correspondence provided in CCA by their time sequence. We use the fixed AE trained on day-0 data to map the neural data of day-0 and day-$k$ onto two sets of $l$-dimensional latent variables, $\{z_0\}$ and $\{z_k\}$, respectively. We then compute the mean and covariance matrices for each of these two empirical distributions, and capture their first and second order statistics by approximating these two distributions by multivariate Gaussians: $p_0(z_0) \sim \mathcal{N}(z_0; \boldsymbol{\mu}_0, \boldsymbol{\Sigma}_0)$ and $p_k(z_k) \sim \mathcal{N}(z_k; \boldsymbol{\mu}_k, \boldsymbol{\Sigma}_k)$. We then minimize the KL divergence between them,

$$D_{\mathrm{KL}}(p_k(z_k)\|p_0(z_0)) = \frac{1}{2}\Big(tr(\boldsymbol{\Sigma}_0^{-1}\boldsymbol{\Sigma}_k) + (\boldsymbol{\mu}_0 - \boldsymbol{\mu}_k)^T\boldsymbol{\Sigma}_0^{-1}(\boldsymbol{\mu}_0 - \boldsymbol{\mu}_k)) - l + ln\frac{|\boldsymbol{\Sigma}_0|}{|\boldsymbol{\Sigma}_k|}\Big) \quad (2)$$

To minimize the KL divergence, we implemented a map from neural activity to latent activity using a network with the same architecture as the encoder section of the BMI's AE. This network was initialized with the weights obtained after training the BMI's AE on the day-0 data. Subsequent training was driven by the gradient of the cost function of equation 2, evaluated on a training set provided by day-$k$ recordings of neural activity. This process aligns the day-$k$ latent PDF to that of day-0 through two global linear operations: a translation through the match of the means, and a rotation through the match of the eigenvectors of the covariance matrices; a nonuniform scaling follows from the match of the eigenvalues of the covariance matrices.

To improve on the Gaussian assumption for the distribution of latent variables, we have trained an alternative BMI in which the AE (Figure 1B) is replaced by a Variational AE (Kingma & Welling, 2013). We train the VAE by adding to the interface loss function (equation 1) a regularizer term: the Kullback-Leibler (KL) divergence $D_{\mathrm{KL}}(p_0(z_0)\|q(z_0))$ between the probability distribution $p_0(z_0)$ of the latent activity on day-0 and $q(z_0) = \mathcal{N}(z_0; 0, I)$. The latent variables of the VAE are thus subject to the additional soft constraint of conforming to a normal distribution.

### 4.2.3 ADVERSARIAL DOMAIN ADAPTATION NETWORK (ADAN)

In addition to matching the probability distributions of latent variables of day-$k$ to those of day-0, we seek an alternative approach: to match the probability distributions of the residuals of the reconstructed firing rates (Zhao et al., 2016), as a proxy for matching the distributions of the neural recordings and their corresponding latent variables. To this end, we train an ADAN whose architecture is very similar to that of a Generative Adversarial Network (GAN): it consists of two deep neural networks, a distribution alignment module and a discriminator module (Figure 1C).

The discriminator is an AE (Zhao et al., 2016) with the same architecture as the one used for the BMI (Figure 1B). The discriminator parameters $\theta_D$ are initialized with the weights of the BMI AE, trained on the day-0 neural data. The goal of the discriminator is to maximize the difference between the neural reconstruction losses of day-$k$ and day-0. The great dissimilarity between the probability distribution of day-0 residuals and that of day-$k$ residuals obtained with the discriminator in its initialized state results in a strong signal that facilitates subsequent discriminator training.

The distribution alignment module works as an adversary to the discriminator by minimizing the neural reconstruction losses of day-$k$ (Warde-Farley & Bengio, 2017). It consists of a hidden layer with exponential units and a linear readout layer, each with $n$ fully connected units. The aligner parameters $\theta_A$, the weights of the $n$ by $n$ connectivity matrices from input to hidden and from hidden to output, are initialized as the corresponding identity matrices. The aligner module receives as inputs the firing rates $X_k$ of day-$k$. During training, the gradients through the discriminator

bring the output $A(\boldsymbol{X}_k)$ of the aligner closer to $\boldsymbol{X}_0$. The adversarial mechanism provided by the discriminator allows us to achieve this alignment in an unsupervised manner.

To train the ADAN, we need to quantify the reconstruction losses. Given input data $\boldsymbol{X}$, the discriminator outputs $\hat{\boldsymbol{X}} = \hat{\boldsymbol{X}}(\boldsymbol{X}, \theta_D)$, with residuals $\boldsymbol{R}(\boldsymbol{X}, \theta_D) = (\boldsymbol{X} - \hat{\boldsymbol{X}}(\boldsymbol{X}, \theta_D))$. Consider the scalar reconstruction losses $\mathbf{r}$ obtained by taking the $L^1$ norm of each column of $\boldsymbol{R}$. Let $\rho_0$ and $\rho_k$ be the distributions of the scalar losses for day-0 and day-$k$, respectively, and let $\mu_0$ and $\mu_k$ be their corresponding means. We measure the dissimilarity between these two distributions by a lower bound to the Wasserstein distance (Arjovsky et al., 2017), provided by the absolute value of the difference between the means: $W(\rho_0, \rho_k) \geq |\mu_0 - \mu_k|$ (Berthelot et al., 2017). The discriminator is trained to learn features that discriminate between the two distributions by maximizing the corresponding Wasserstein distance. The discriminator initialization implies $\mu_k > \mu_0$ when the ADAN training begins. By maximizing $(\mu_k - \mu_0)$, equivalent to minimizing $(\mu_0 - \mu_k)$, this relation is maintained during training. Since scalar residuals and their means are nonnegative, the maximization of $W(\rho_0, \rho_k)$ is achieved by decreasing $\mu_0$ while increasing $\mu_k$.

Given discriminator and aligner parameters $\theta_D$ and $\theta_A$, respectively, the discriminator and aligner loss functions $\mathcal{L}_D$ and $\mathcal{L}_A$ to be minimized can be expressed as

$$\begin{cases} \mathcal{L}_D = \mu_0(\boldsymbol{X}_0; \theta_D) - \mu_k(A(\boldsymbol{X}_k; \theta_A); \theta_D) & \text{for } \theta_D \\ \mathcal{L}_A = \mu_k(A(\boldsymbol{X}_k; \theta_A); \theta_D) & \text{for } \theta_A \end{cases} \tag{3}$$

## 5 RESULTS

Figure 2A illustrates the firing rates ($n = 96$), the latent variables ($l = 10$), the reconstructed firing rates, and the actual and predicted muscle activity for two representative muscles, a wrist flexor and an extensor, over a set of eight trials (one trial per target location) of a test data set randomly selected from day-0 data. The overall performance of the interface is summarized in Figure 2B, quantified using the percentage of the variance accounted for (%VAF) for five-fold cross-validated data. The blue bar shows the accuracy of EMG predictions directly obtained from the $n$-dimensional firing rates, without the dimensionality reduction step; this EMG predictor consists of an LSTM layer with $n$ units, followed by a linear layer. The green bar shows the accuracy of EMG predictions when the interface is trained sequentially: first the AE is trained in an unsupervised manner, and then the EMG predictor is trained with the resulting $l$-dimensional latent variables as input. The performance of the sequentially trained interface is worse than that of an EMG predictor trained directly on neural activity as input; the difference is small but significant (paired t-test, p=0.006). In contrast, when the EMG predictor is trained simultaneously with the AE (red bar), there is no significant difference (paired t-test, p=0.971) in performance between EMG predictions based on the $n$-dimensional neural activity and the $l$-dimensional latent variables. In simultaneous training, the AE is trained using the joint loss function of equation 1 that includes not only the unsupervised neural reconstruction loss but also a supervised regression loss that quantifies the quality of EMG predictions. Therefore, the supervision of the dimensionality reduction step through the integration of relevant movement information leads to a latent representation that better captures neural variability related to movement intent.

For the implementation of the domain adaptation techniques, the interface was trained using only the data of day-0 and remained fixed afterward. Both CCA and KLDM were designed to match latent variables across days. Therefore, when using these methods, we first use the encoder part of the fixed AE to map neural activity of subsequent days onto latent activity, we then apply CCA or KLDM to align day-$k$ latent activity to that of day-0, and finally use the fixed EMG predictor to predict EMGs from the aligned latent activity. While KLDM explicitly seeks to match first and second order statistics of the latent variables through a Gaussian approximation, CCA aligns the latent variables using a point-to point correspondence across days provided by the latent trajectories. The effect of CCA alignment is illustrated in Figure 3A, where we show 2D t-SNE visualizations of 10D latent trajectories. Each trajectory is an average over all trials for a given target. The differences between day-16 and day-0 latent trajectories reflect the impact of turnover in the recorded neurons. Comparison between these two sets of trajectories reveals a variety of transformations, including nonuniform rotation, scaling, and skewing. In spite of the complexity of these transformations, the available point-to-point correspondence along these trajectories allows CCA to achieve a good alignment. The mechanism underlying KLDM alignment is illustrated in 3B, where we show the

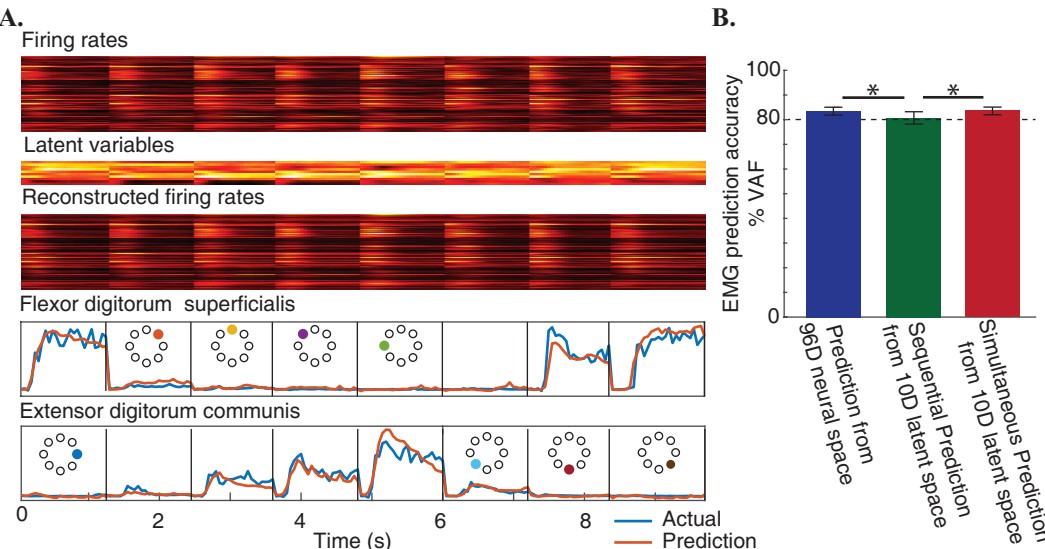

Figure 2: Neural to muscle BMI. **A.** Example firing rates recorded from the hand area of primary motor cortex while the monkey performed the isometric wrist task; we also show latent variables and reconstructed firing rates. The quality of EMG predictions is illustrated by comparison to actual EMGs for two representative muscles, for each of the eight target directions. **B.** Performance comparison between EMG predictions from $n$-dimensional firing rates (blue) and EMG predictions from $l$-dimensional latent variables, obtained either by training the predictor sequentially (green) or simultaneously (red) with the neural AE. Error bars represent standard deviation of the mean.

empirical probability distribution of the latent variables along a randomly chosen, representative direction within the 10D latent space. Results are shown for day-0 (blue), for day-16 (red), and for day-16 after alignment with KLDM (yellow). The effects of using a BMI based on a VAE instead of the AE are shown in Supplementary Figure S1.

In contrast to CCA and KLDM, ADAN is designed to match the high-dimensional neural recordings across days via the $L^1$ norm of their residuals. In the ADAN architecture, the discriminator module is an AE that receives as inputs the neural activity of day-0 or day-$k$ and outputs their reconstructions. The residuals of the reconstruction of neural signals follow from the difference between the discriminator's inputs and outputs. The ADAN aligns the day-$k$ residual statistics to those of day-0 by focusing on the $L^1$ norm of the residuals and minimizing the distance between these scalar PDFs across days; this procedure results in the alignment of the neural recordings and consequently their latent representation. The aligner module of ADAN aligns day-$k$ neural activity to that of day-0; this aligned neural activity is then used as input to the fixed BMI. Figure 3C shows the 1D distribution of the $L^1$ norm of the vector residuals, and in Figure 3D a 2D t-SNE (Maaten & Hinton, 2008) visualization of the vector residuals based on 1000 randomly sampled data points. Residuals correspond to the errors in firing rate reconstructions using the day-0 fixed AE for both day-0 data (blue) and day-16 data (red). Residuals for the day-16 data after alignment with ADAN are shown in yellow. Figure 3E shows the empirical probability distribution of the latent variables along the same representative, randomly chosen dimension within the 10D latent space used in Figure 3B. Results are shown for latent variables on day-0 using the fixed AE (blue), for the latent variables on day-16 along the same dimension using the same, fixed AE (red), and for day-16 latent variables after alignment with ADAN (yellow). A 2D t-SNE visualization of latent variables aligned with ADAN is shown in comparison to the results of a simple center-and-scale alignment in Supplementary Figure S2.

The performance of the BMI before and after domain adaptation with CCA, KLDM, and ADAN is summarized in Figure 4A and quantified using the %VAF in EMG predictions. We report mean and standard deviation for five-fold cross-validated data. Blue symbols indicate the performance of an interface that is updated on each day; this provides an upper bound for the potential benefits

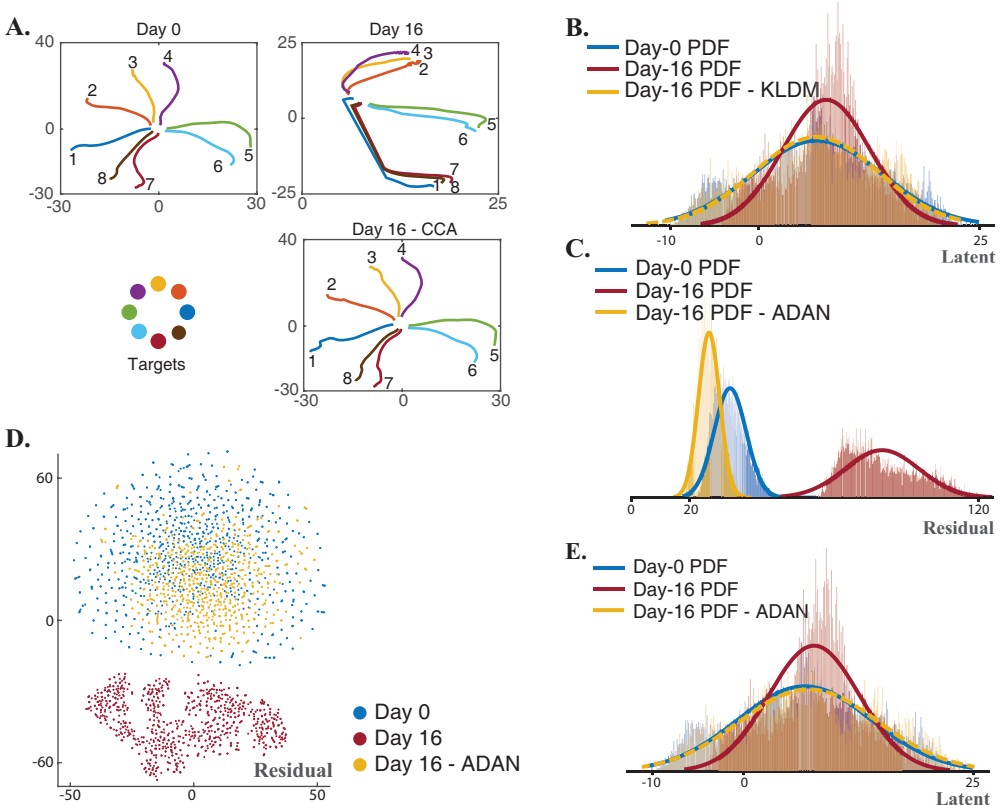

Figure 3: Domain adaptation. **A.** 2D t-SNE visualization of the averaged 10D latent neural trajectories as the monkey performed the isometric wrist task for day-0, day-16 before alignment, and day-16 after alignment with CCA. **B.** Probability distribution of the 10D latent variables along a randomly chosen, representative direction. We show the distribution at day-0, and the distribution at day-16 before and after alignment using KLDM. **C.** Probability distribution of the $L^1$ norm of the vector residuals of the reconstructed firing rates for day-0 and day-16, before and after adversarial alignment using ADAN. **D.** 2D t-SNE visualization of the vector residuals of the reconstructed firing rates for day-0 and day-16, before and after adversarial alignment using ADAN. **E.** Same as **B**, but for alignment using ADAN.

of neural domain adaptation. Red symbols illustrate the natural deterioration in the performance of a fixed interface due to the gradual deterioration of neural recordings. Green, orange, and purple symbols indicate the extent to which the performance of a fixed interface improves after alignment using CCA, KLDM, and ADAN, respectively. The comparable performance of CCA and KLDM reflects that both methods achieve alignment based on latent statistics; the use of ADAN directly for latent space alignment does not produce better results than these two methods. In contrast, when ADAN is used for alignment based on residual statistics, interface stability improves. This ADAN provides a better alignment because the residuals amplify the mismatch that results when a fixed day-0 AE is applied to later-day data (see Figures 3C and D). Although the improvement achieved for day-16 with ADAN over its competitors is small, about $6\%$, it is statistically significant (one-way ANOVA with Tukey's test, $p < 0.01$). We have been unable to achieve this degree of improvement with any of the many other domain adaptation approaches we tried. This improvement is even more remarkable given that domain adaptation with ADAN requires a surprisingly small amount of data. Figure 4B shows the percentage improvement in EMG predictions as a function of the amount of training data. Subsequent symbols are obtained by adding 6s of data (120 samples) to the training set, and computing the average percentage improvement for the entire day (20 min recordings), for all days after day-0. EMG prediction accuracy saturates at ~1 min; this need for a small training set is ideal for practical applications.

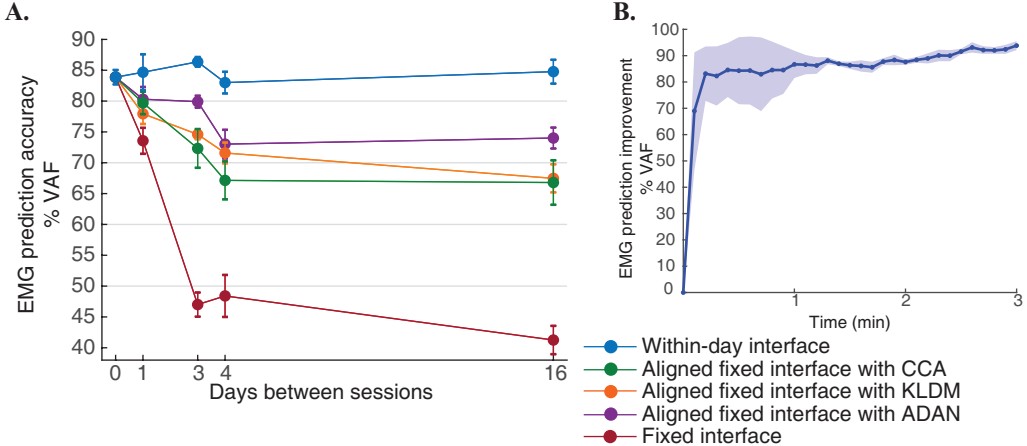

Figure 4: **A.** EMG prediction performance using a fixed BMI decoder. Blue symbols represent the sustained performance of interfaces retrained on a daily basis. Red symbols illustrate the deterioration in performance of a fixed interface without domain adaptation. The performance of a fixed interface after domain adaptation is shown for CCA (green), KLDM (orange), and ADAN (purple). Error bars represent standard deviation of the mean. **B.** Average improvements in EMG prediction performance for alignment using ADAN as a function of the amount of training data needed for domain adaptation at the beginning of each day, averaged over all days after day-0. Shading represents standard deviation of the mean.

## 6 CONCLUSION

We address the problem of stabilizing a fixed Brain-Machine Interface against performance deterioration due to the loss and turnover of recorded neural signals. We introduce a new approach to extracting a low-dimensional latent representation of the neural signals while simultaneously inferring movement intent. We then implement various domain adaption methods to stabilize the latent representation over time, including Canonical Correlation Analysis and the minimization of a Kullback-Leibler divergence. These two methods provide comparable improvement in the performance of the interface. We find that an Adversarial Domain Adaptation Network trained to match the empirical probability distribution of the residuals of the reconstructed neural recordings restores the latent representation of neural trajectories and outperforms the two methods based on latent variables, while requiring remarkably little data to solve the domain adaptation problem. In addition, ADAN solves the domain adaptation problem in a manner that is not task specific, and thus is potentially applicable to unconstrained movements.

Here we report on improvements in interface stability obtained offline, without a user in the loop. Online, closed-loop performance is not particularly well correlated with offline accuracy; in an online evaluation of performance, the user's ability to adapt at an unknown rate and to an unknown extent to an imperfect BMI obscures the performance improvements obtained with domain adaptation. Although the open-loop performance improvement demonstrated here is encouraging, additional experiments, both open and closed-loop, with additional animals and involving additional tasks, are required to fully validate our results and to establish that the improvements demonstrated here facilitate the sustained use of a brain-machine interface.

ACKNOWLEDGMENTS

These results are based upon work supported the National Institute of Neurological Disorders and Stroke (NINDS) under Grant No. 5R01NS053603-12. We thank F. Mussa-Ivaldi, E. Perreault, X. Ma, K. Bodkin, E. Altan, and M. Medina for discussions and comments.

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

SUPPLEMENTARY MATERIAL

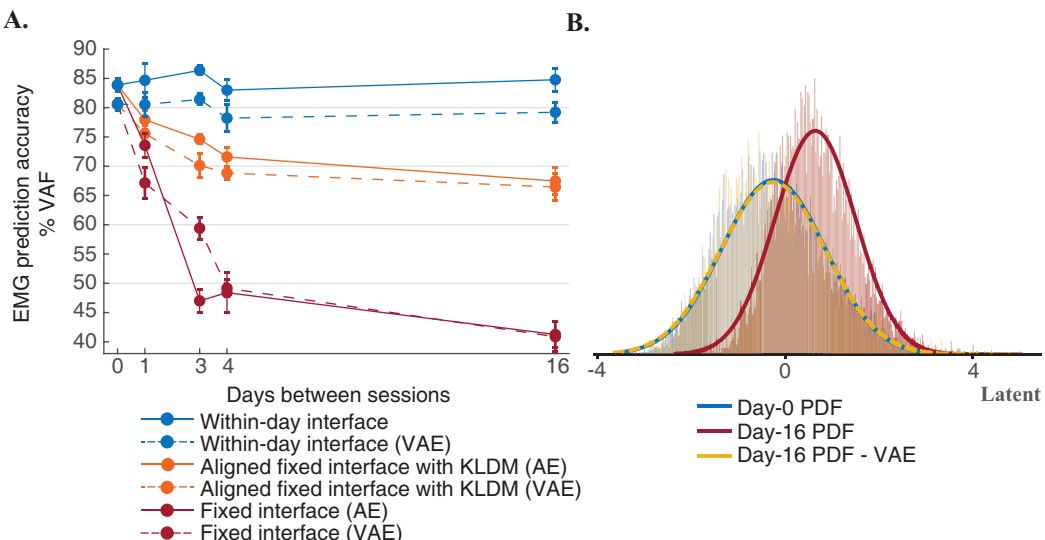

Figure S1: Variational autoencoder. **A.** EMG prediction performance using a different BMI, based on a VAE decoder, in comparison to the performance of a BMI based on the traditional autoencoder. Blue symbols represent the sustained performance of an interface retrained on a daily basis. Red symbols illustrate the deterioration in the performance of a fixed interface without domain adaptation. Orange symbols represent the performance of a fixed interface when the latent variables are aligned using KLDM. For each of these three cases, solid lines represent the performance of an AE-based BMI, and dashed lines that of a VAE-based BMI. Error bars represent standard deviation of the mean. **B.** Probability distribution of the 10D latent variables along the same dimension used in Figure 3B, now obtained with the fixed VAE trained on day-0. We show the distribution at day-0, and the distribution at day-16 before and after alignment using KLDM. In comparison to Figure 3B, the use of a VAE greatly improves the Gaussian nature of the latent variables' distribution. However, this additional constraint in the autoencoder results in a slight deterioration of the BMI's ability to predict EMGs, as shown in **A.**

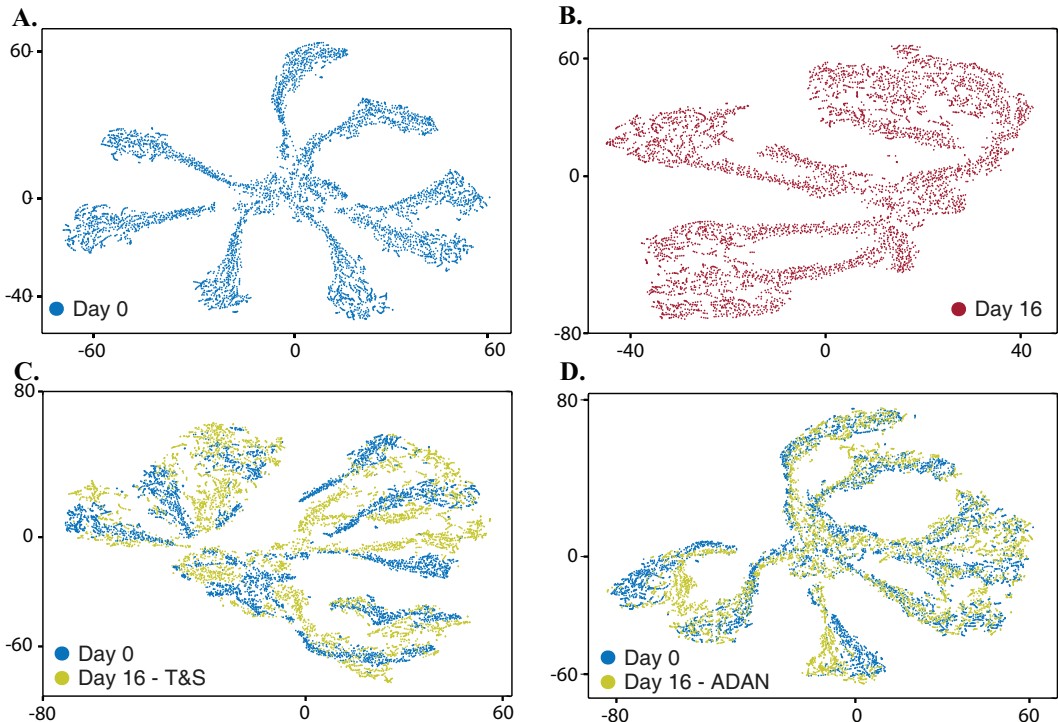

Figure S2: Visualization of the probability distribution of latent variables in 2D using t-SNE. **A.** Latent variables on day-0. **B.** Latent variables on day-16, before alignment. **C.** Latent variables on day-16, after alignment to those of day-0 using T&S: a global translation to match the respective means followed by a global scaling to match the respective variances (yellow). Also shown, latent variables on day-0 (blue) on the same projection. **D.** Latent variables on day-16, after alignment to those of day-0 using ADAN (yellow). Also shown, latent variables on day-0 (blue) on the same projection.

