# OpenReview forum: "Adversarial Domain Adaptation for Stable Brain-Machine Interfaces"
_ICLR.cc/2019/Conference_

### Official Review · AnonReviewer2 · 2018-10-20
**ways to deal with nonstationarity in BCIs**

**Rating:** 7
**Confidence:** 5

**Review:**

The paper considers invasive BMIs and studies various ways to avoid daily recalibration due to changes in the brain signals.
While I like the paper and studied methods -- using adverserial domain adaptation is interesting to use in this context --, I think that the authors oversell a bit.
The problem of nonstationarity rsp. stability is an old one in non-invasive BCIs (shenoy et al JNE 2006 was among the first) and a large number of prior methods have been defined to robustify feature spaces, to project to stable subspaces etc. Clearly no Gans at that time. The least the authors could do is to make reference to this literature, some methods may even apply also for the invasive data of the paper.
While the authors did not clearly say that they present an offline analysis; one method, the GAN, gets 6% better results then the competitors. I am not sure whether this is practically relevant in an online setting. But this needs to be clearly discussed in the paper and put into perspective  to avoid wrong impression. Only an online study would be convincing.

Overall, I think the paper could be accepted, the experiments are nice, the data is interesting, if it is appropriately toned down (avoiding statements about having done something for the first time) and properly references to prior work are given. It is an interesting application domain. I additionally recommend releasing the data upon acceptance.

---

> ### Author Response · Authors · 2018-11-17
> **Response to Reviewer 2**
>
> We thank the reviewer for the feedback and comments.
>
> Q: “The paper considers invasive BMIs and studies various ways to avoid daily recalibration due to changes in the brain signals. While I like the paper and studied methods -- using adversarial domain adaptation is interesting to use in this context --, I think that the authors oversell a bit. The problem of nonstationarity rsp. stability is an old one in non-invasive BCIs (shenoy et al JNE 2006 was among the first) and a large number of prior methods have been defined to robustify feature spaces, to project to stable subspaces etc. Clearly no Gans at that time. The least the authors could do is to make reference to this literature, some methods may even apply also for the invasive data of the paper.”
>
> A: We thank the reviewer for the positive comment about our work. We do not claim to be the first to address the issue of stability in the presence of non-stationary recorded signals. We have added references to Zhang & Chase 2013, Nuyujukian et al 2014, Dyer et al 2017, and Downey et al 2018 to the papers we had already listed in the section on Related Work: Orsborn et al 2012, Dangi et al 2013, Bishop et al 2014, Jarosiewicz et al 2015, Susillo et al 2016, Kao et al 2017, and Pandarinath et al 2017. We could not find the Shenoy et al JNE 2006 article mentioned by the reviewer. We would appreciate more details regarding this paper – does this refer to the Nature 2006 or the JNE 2007 paper from the Shenoy group?  In the revised version of our paper, we have expanded the description of the methods previously put forward by these authors. Our claim to novelty is in formulating the problem as one of domain adaptation for neural signals, a problem that can be addressed through the use of adversarial training.
>
> Q: “While the authors did not clearly say that they present an offline analysis; one method, the GAN, gets 6% better results then the competitors. I am not sure whether this is practically relevant in an online setting. But this needs to be clearly discussed in the paper and put into perspective to avoid wrong impression. Only an online study would be convincing.”
>
> A: The reviewer is correct in pointing out that our evaluation of BMI performance is offline, in an open-loop scenario and that we cannot claim improved ease of use since the aligned BMI has not been tested in an online, closed-loop scenario. We have removed the statement to that effect in the revised version of the paper. However, the 6% improvement over the competitors in open-loop was statistically significant. A further advantage of ADAN in comparison to CCA and KLDM is that it is an unsupervised method that involves no assumption on the statistics of the latent activity.
> The question of online vs offline comparison is an important one. Online BMI performance is not perfectly correlated with offline accuracy; this is actually the reason that offline comparison is important in this case. In an online evaluation of BMI performance, the user’s ability to adapt at an unknown rate and to an unknown extent to an imperfect BMI obscures the performance improvements obtained with domain adaptation.  Although experiments, both open and closed loop, with additional animals and involving additional tasks, are in process as required to validate our results, the open loop performance improvement demonstrated here is a more stringent metric than improvements achieved in a closed loop setting.
>
> Q: “Overall, I think the paper could be accepted, the experiments are nice, the data is interesting, if it is appropriately toned down (avoiding statements about having done something for the first time) and properly references to prior work are given. It is an interesting application domain. I additionally recommend releasing the data upon acceptance.”
>
> A: We once more thank the reviewer for the positive comments. We have followed the advice and are more careful and specific in claiming novelty in the revised version of the paper. We plan to make data and code available on GitHub upon acceptance.

---

### Official Review · AnonReviewer1 · 2018-10-25
**Review for Adversarial Domain Adaptation for Stable Brain-Machine Interfaces**

**Rating:** 5
**Confidence:** 3

**Review:**

Here the authors define a BMI that uses an autoencoder -> LSTM -> EMG. The authors then address the problem of data drift in BMI and describe a number of domain adaptation algorithms from simple (CCA to more complex ADAN) to help ameliorate it. There are a lot of extremely interesting ideas in this paper, but the paper is not particularly well written, and the overall effect to me was confusion. What problem is being solved here? Are we describing using latent variables (AE approach) for BMI?  Are we discussing domain adaptation, i.e. handling the nonstationarity that so plagues BMI and array data?  Clearly the issue of stability is being addressed but how?  A number of different approaches are described from creating a pre-execution calibration routine whereby trials on the given day are used to calibrate to an already trained BMI (e.g. required for CCA) to putting data into an adversarial network trained on data from earlier days.  Are we instead attempting to show that a single BMI can be used across multiple days?


This paper is extremely interesting but suffers from lack of focus, rigor, and clarity.
Focus :
AE to RNN to EMG is that the idea to compare vs. Domain adaptation via CCA/KLDM/ADAM.
Of course a paper can explore multiple ideas, but in this case the comparisons and controls for both are not adequate.

Rigor:
What are meaningful comparisons for all for the AE and DA portions? The AE part is strongly related to either to Kao 2017 or Pandarinath 2018 but nothing like that is compared.  The domain adaptation part evokes data augmentation strategies of Sussillo 2016 but that is not compared.

If I were reviewing this manuscript for a biological journal a rigorous standard would be online BMI results in two animals.  Is there a reason why this isn’t the standard for ICLR? Is the idea that non-biological journals / conferences are adequate to vet new ideas before really putting them to the test in a biological journal?  The manuscript is concerned with the vexing problem of BMI stability of time, which seems to be a problem where online testing in two animals would be critical. (I appreciate this is a broader topic relevant to the BMI field beyond just this paper, but it would be helpful to get some thinking on this in the rebuttal).

Clarity :
This paper needs to be pretty seriously clarified.  The mathematical notation is not adequate to the job, nor is the motivation for the varied methodology. I cannot tell if the subscript is for time or for day. Also, what is the difference between z_0 vs. Z_0? I do not know what exactly is going into the AE or the ADAN.

The neural networks are not described to a point where one could reproduce this work. The notation for handling time is inadequate.   E.g. despite repeated readings I cannot tell how time is handled in the auto-encoder, e.g. nxt is vectorized vs feeding n-sized vector one time step at a time?


Questions

What is the point of the latent representation in the AE if it is just fed to an LSTM? Is it to compare to not using it?

Page 3, how precisely is time handled in the AE?  If time is just vectorized, how can one get real-time readouts? In general there is not enough detail to understand what is implemented in the AE. If only one time slice is entered into AE, then it seems clear AE won’t be very good because one desires latent representation of the dynamics, not single time slices.

How big is the LSTM used to generate the EMG?

It seems like a the most relevant baseline is to compare to the data perturbation strategies in Sussillo 2016.  If you have an LSTM already up and running to predict EMG, this seems very doable.

Page 4, “We then use an ADAN to align either the distribution of latent variables or the distributions of the residuals of the reconstructed neural data, the latter a proxy for the alignment of the neural latent variables.”  This sentence is not adequate to explain the concepts of the various distributions, the residuals of reconstructed neural data (where do the residuals come from?), and why is one a proxy for the other.  Please expand this sentence into a few sentences, if necessary to define these concepts for the naive reader.

Page 5, What parameters are minimized in equation (2)? Please expand the top sentence of page 5.

Page 6, top - “In contrast, when the EMG predictor is  trained simultaneously with the AE…” Do you mean there is again a loss function defined by both EMG prediction and AE and summed, and then backprop is used to train both in an end-to-end fashion?  Please clarify.

Page 8, How do the AE results and architecture fit into the EMG reconstruction “BMI” results? Is that all decoding results are first put through the AE -> LSTM -> EMG pipeline? I.e. your BMI is neural data -> AE -> LSTM -> EMG?  If so, then how does the ADAN / CCA and KLDM fit in?  You first run those three DA algorithms and then pipe it through the BMI?

Page 8, How can you say that the BMI improvement of 6% is meaningful to the BMI user if you did not test the BMI online?

---

> ### Author Response · Authors · 2018-11-14
> **Response to Reviewer 1 (4/4)**
>
> Q: “Page 5, What parameters are minimized in equation (2)? Please expand the top sentence of page 5.”
>
> A: The KLD of Eq 2 is always positive, and reaches its minimum at zero when the mean and covariance matrix for day-k match those for day-0. The KLDM method thus aligns the latent statistics of day-k to those of day-0 by implementing a transformation that equalizes the first and second moments of these complex PDFs.  To minimize the KLD, we used a map from neural activity to latent activity implemented by a network with the same architecture as the encoder section of the BMI’s AE. This network was initialized with the weights obtained after training the BMI’s AE on the day-0 data. Training proceeded on inputs provided by day-k recordings of neural activity. The loss function on the latent variables was as shown in Eq 2. We have added these clarifications to the revised version of our paper.
>
> Q: “Page 6, top - “In contrast, when the EMG predictor is trained simultaneously with the AE…” Do you mean there is again a loss function defined by both EMG prediction and AE and summed, and then backprop is used to train both in an end-to-end fashion? Please clarify.”
>
> A: When the EMG predictor is trained simultaneously with the AE, the AE is trained using the joint loss function of Eq 1.  The alternative, is to independently train the AE in a purely unsupervised manner, not including the second term in Eq 1. We have clarified this point in the revised version of our paper.
>
> Q: “Page 8, How do the AE results and architecture fit into the EMG reconstruction “BMI” results? Is that all decoding results are first put through the AE -> LSTM -> EMG pipeline? I.e. your BMI is neural data -> AE -> LSTM -> EMG? If so, then how does the ADAN / CCA and KLDM fit in? You first run those three DA algorithms and then pipe it through the BMI?”
>
> A: The BMI consists of two computational modules: the neural AE and the EMG predictor. These were trained using only the data of day-0 and remained fixed afterward. Once the BMI is trained, the fixed encoder part of the AE maps neural activity into latent activity. Both CCA and KLDM were designed to match latent variables across days. Therefore, when using these methods, we first obtained the latent variables Zk of subsequent days using the encoder part of the fixed AE of the BMI, then applied CCA and KLDM to align these latent variables to those of day-0, and finally used the fixed EMG predictor to predict EMGs from the aligned latent variables. In contrast, ADAN was designed to match high-dimensional neural recordings across days.  Therefore, when using ADAN, first we aligned the neural recordings Xk of a subsequent day to those of a day-0 and then used the aligned vectors of neural activity as inputs to the fixed BMI. We have clarified this aspect of domain adaptation in the revised version of our paper.
>
> Q: “Page 8, How can you say that the BMI improvement of 6% is meaningful to the BMI user if you did not test the BMI online?”
>
> A: We agree. We have removed this sentence from the revised version of our paper.
>
> We thank the reviewer again for the feedback and comments, which have improved the manuscript.

---

> ### Author Response · Authors · 2018-11-15
> **Response to Reviewer 1 (3/4)**
>
> Q: “What is the point of the latent representation in the AE if it is just fed to an LSTM? Is it to compare to not using it?”
>
> A: The high degree of correlation in the activity of M1 neurons makes the use of dimensionality reduction methods a common practice in BMI design. Expected advantages are the denoising of the neural recordings and the possibility of using a more compact representation of neural activity as input to the predictor of muscle activity. Here we proposed an approach to AE training that results in a latent space based muscle predictor that performs as well as a muscle predictor based directly on the high dimensional neural activity.
>
> Q: “Page 3, how precisely is time handled in the AE? If time is just vectorized, how can one get real-time readouts? In general there is not enough detail to understand what is implemented in the AE. If only one time slice is entered into AE, then it seems clear AE won’t be very good because one desires latent representation of the dynamics, not single time slices.”
>
> A: The AE is trained using batch training on a loss function that accumulates in additive form the loss associated with each individual training example. The AE is trained to produce an optimal reconstruction of the neural vectors xt regardless of their temporal order. After training, real-time readouts for the latent activity can be obtained for every successively presented neural activity input. If the goal is to track dynamics in latent space, latent trajectories can be constructed by concatenating the latent representations in the appropriate order, as when using CCA. The other two domain adaptation techniques that we implemented, KLDM and ADAN, do not require this concatenation, as they focus on matching the statistics of the latent variables as opposed to their dynamics.
>
> Q: “How big is the LSTM used to generate the EMG?”
>
> A: The number of units in the LSTM layer is equal to the number of the recorded muscles (m = 14). We have added this information in the revised paper.
>
> Q: “It seems like a the most relevant baseline is to compare to the data perturbation strategies in Sussillo 2016. If you have an LSTM already up and running to predict EMG, this seems very doable.”
>
> A: The data perturbations applied in Sussillo 2016 include electrode dropping or changing the average firing rate of individual neurons. The actual neural turnover results in complex transformations of the latent activity that cannot be simulated using these simple data perturbations. The high degree of correlation in the firing activity of M1 neurons implies that dropping individual channels should not have a large impact on the BMI’s performance, an expectation supported by our own preliminary analysis, unpublished. Moreover, our analysis shows that an alignment method that only compensates for translation (i.e. by changing individual average firing rates) and scaling perturbations, would fail to implement the complex transformations needed to match latent distributions (see Fig S2).
>
> Q: “Page 4, “We then use an ADAN to align either the distribution of latent variables or the distributions of the residuals of the reconstructed neural data, the latter a proxy for the alignment of the neural latent variables.” This sentence is not adequate to explain the concepts of the various distributions, the residuals of reconstructed neural data (where do the residuals come from?), and why is one a proxy for the other. Please expand this sentence into a few sentences, if necessary to define these concepts for the naive reader.”
>
> A: Initially, we used an adversarial network to directly match the PDF of the latent variables across days; the resulting improvements in EMG prediction with this approach to domain adaptation were comparable to those obtained with KLDM and CCA. Next, we implemented ADAN to match PDFs in neural space; not the PDF of the reconstructed neural activity but that of the L1 norm of their residuals (the difference between actual and reconstructed neural activity). In the ADAN architecture, the discriminator is an AE that receives as inputs the neural activity of day-0 and day-k and outputs their reconstructions. The residuals follow from the difference between the discriminator’s inputs and outputs. The ADAN aligns the day-k statistics to those of day-0 by minimizing the distance between the PDFs of the respective scalar residuals. This procedure results in the alignment of the neural recordings and consequently their latent representation across days (Fig 3 C-E). We have expanded this sentence and clarified these points in the revised version of our paper.

---

> ### Author Response · Authors · 2018-11-15
> **Response to Reviewer 1 (2/4)**
>
> Q: “What are meaningful comparisons for all for the AE and DA portions? The AE part is strongly related to either to Kao 2017 or Pandarinath 2018 but nothing like that is compared. The domain adaptation part evokes data augmentation strategies of Sussillo 2016 but that is not compared.”
>
> A: The use of dimensionality reduction to design BMIs is not novel. This approach has been triggered by the observation of a high degree of correlation in the activity of individual M1 neurons, the expectation of obtaining a more compact and denoised representation of neural activity, and the convenience of using a low-dimensional signal as input to the EMG predictor to simplify its training and avoid overfitting.  Most of the earlier work used linear dimensionality reduction methods such as PCA and FA to obtain the latent variables (e.g. Yu et al., 2009; Shenoy et al., 2013; Sadtler et al., 2014; Gallego et al., 2017a). More recently, the use of AEs as a nonlinear dimensionality reduction method has been investigated by Pandarinath et al. (2018). Our contribution here, as discussed above, is to combine unsupervised and supervised goals in the AE training, and to show that this results in improved BMI performance. We have clarified this point in the revised version of the paper.
> There is not much previous work on the question of stabilizing BMI performance against neural turnover. In Sussillo 2016, the authors used months of recordings to train a BMI and to make it robust to neural changes. Here, we seek to find methods that allow us to stabilize the BMI using single session data. Data augmentation strategies and domain adaptation techniques are inherently different but complementary approaches.
>
> Q:” If I were reviewing this manuscript for a biological journal a rigorous standard would be online BMI results in two animals. Is there a reason why this isn’t the standard for ICLR? Is the idea that non-biological journals / conferences are adequate to vet new ideas before really putting them to the test in a biological journal? The manuscript is concerned with the vexing problem of BMI stability of time, which seems to be a problem where online testing in two animals would be critical. (I appreciate this is a broader topic relevant to the BMI field beyond just this paper, but it would be helpful to get some thinking on this in the rebuttal).”
>
> A: The question of online vs offline comparison is an important one. Online BMI performance is not perfectly correlated with offline decoder accuracy; this is actually the reason that offline comparison is important in this case. In an online evaluation of BMI performance, the user’s ability to adapt at an unknown rate and to an unknown extent to an imperfect BMI obscures the performance improvements obtained with domain adaptation.  We do agree that additional experiments, both open and closed loop, with additional animals and involving additional tasks, are required to fully validate our results; we are currently in the process of developing and running these experiments. A vetting of the computational ideas within the machine learning community is invaluable before implementing closed-loop experiments.
>
> Q: “This paper needs to be pretty seriously clarified. The mathematical notation is not adequate to the job, nor is the motivation for the varied methodology. I cannot tell if the subscript is for time or for day. Also, what is the difference between z_0 vs. Z_0? I do not know what exactly is going into the AE or the ADAN.
>
> A: The subscript t in Eq 1 labels the time ordered data points that constitute the training set on day-0. This has been further clarified in the paper. As indicated in the original version, day-k is the notation adopted to indicate successive days following day-0. Capital letters are used to represent matrices: X0 and Xk are n by T matrices that aggregate the neural data for day-0 and day-k, respectively; while Z0 and Zk are l by 8τ matrices (as explained in the paper) that aggregate the latent activity for day-0 and day-k, respectively. Lowercase letters represent vectors, with x referring to neural activity, z to latent activity, and y to muscle activity. The inputs to the BMI and the ADAN are neural recordings, as shown in Fig 1.
>
> Q: “The neural networks are not described to a point where one could reproduce this work. The notation for handling time is inadequate. E.g. despite repeated readings I cannot tell how time is handled in the auto-encoder, e.g. nxt is vectorized vs feeding n-sized vector one time step at a time?”
>
> A: As indicated in Eq 1, the training of the AE was guided by a loss function that is a sum over the loss for each input vector xt. Each n-dimensional input vector, labeled by t, is individually fed to the BMI to obtain the corresponding neural activity reconstruction and muscle activity prediction. The corresponding loss is calculated and combined additively to compute a cumulative gradient used for batch training.

---

> ### Author Response · Authors · 2018-11-15
> **Response to Reviewer 1 (1/4)**
>
> We thank the reviewer for the detailed feedback and comments.
>
> Q: “Here the authors define a BMI that uses an autoencoder -> LSTM -> EMG. The authors then address the problem of data drift in BMI and describe a number of domain adaptation algorithms from simple (CCA to more complex ADAN) to help ameliorate it. There are a lot of extremely interesting ideas in this paper, but the paper is not particularly well written, and the overall effect to me was confusion. What problem is being solved here? Are we describing using latent variables (AE approach) for BMI? Are we discussing domain adaptation, i.e. handling the nonstationarity that so plagues BMI and array data? Clearly the issue of stability is being addressed but how? A number of different approaches are described from creating a pre-execution calibration routine whereby trials on the given day are used to calibrate to an already trained BMI (e.g. required for CCA) to putting data into an adversarial network trained on data from earlier days. Are we instead attempting to show that a single BMI can be used across multiple days?”
>
> A: The reviewer correctly highlights a number of the important aspects of the work. Our main objective is indeed to stabilize a fixed BMI so as to increase its longevity and usability across many days. To this end, we started by describing the general architecture and training of the BMI, which is trained on day-0 and consists of two components. The first is an autoencoder that provides a nonlinear map from neural signals to a low dimensional space of latent signals. The second is an EMG predictor that maps the latent signals onto muscle activity. The first point of our paper is that better BMI performance is achieved when the training of the AE is based on a loss function (see Eq 1) that includes not only the unsupervised neural reconstruction loss but also a supervised regression loss that quantifies the quality of EMG prediction.
> The goal is to keep this BMI fixed so that the user only needs to adapt to it once. However, the performance of a fixed BMI will deteriorate because of neural turnover (see blue data on Fig 3A). A way to maintain performance in the face of changing neural signals is to keep on retraining the interface (see red data on Fig 3A), but this is not a viable solution as it requires the user to keep on adapting to a new interface on an almost daily basis. We have expanded in the revised version of our paper on the problems caused by frequent BMI recalibration. To avoid this problem, our approach was to investigate interventions on the latent space representations to align and stabilize the inputs to the EMG predictor without changing the AE. To this end, we explored three domain adaptation approaches and showed that the use of ADAN provides the most effective solution.
>
> Q: “AE to RNN to EMG is that the idea to compare vs. Domain adaptation via CCA/KLDM/ADAM. Of course a paper can explore multiple ideas, but in this case the comparisons and controls for both are not adequate.”
>
> A: As explained above, we first described the architecture and training of the BMI. We trained and fixed this BMI on the data of day-0. In the loss function of Eq 1, used to train the AE on day-0, the index t from 0 to T labels the day-0 data. Each input to the BMI is an n-dimensional vector x  of neural data.  The performance of this BMI, kept fixed, quickly deteriorates due to neural turnover. We implemented three domain adaptation methods (CCA, KLDM, and ADAN) to stabilize the performance of the fixed BMI across subsequent days and identified which domain adaptation technique provides the most stability to a fixed BMI.

---

> ### Author Response · Authors · 2018-11-21
> **Revised Version**
>
> Dear reviewer #1,
>
> We have submitted a response to your review and a revised version of our paper.  We hope to have succeeded in answering all questions and comments. If there are any remaining concerns, please let us know so that we can address them before the deadline.
>
> Thanks,

---

### Official Review · AnonReviewer3 · 2018-11-06
**Review of adversarial domain adaptation for stable brain-machine interfaces**

**Rating:** 9
**Confidence:** 4

**Review:**

This contribution describes a novel approach for implanted brain-machine interface in order to address calibration problem and covariate shift. A latent representation is extracted from SEEG signals and is the input of a LTSM trained to predict muscle activity. To mitigate the variation of neural activities across days, the authors compare a CCA approach, a Kullback-Leibler divergence minimization and a novel adversarial approach called ADAN.

The authors evaluate their approach on 16-days recording of neurons from the motor cortex of rhesus monkey, along with EMG recording of corresponding the arm and hand. The results show that the domain adaptation from the first recording is best handled with the proposed adversarial scheme. Compared to CCA-based and KL-based approaches, the ADAN scheme is able to significantly improve the EMG prediction, requiring a relatively small calibration dataset.

The individual variability in day-to-day brain signal is difficult to harness and this work offers an interesting approach to address this problem. The contributions are well described, the limitation of CCA and KL are convincing and are supported by the experimental results. The important work on the figure help to provide a good understanding of the benefit of this approach.

Some parts could be improved. The results of Fig. 2B to investigate the role of latent variables extracted from the trained autoencoder are not clear, the simultaneous training could be better explained. As the authors claimed that their method allows to make an unsupervised alignment neural recording, independently of the task, an experiment on another dataset could enforce this claim.

---

> ### Author Response · Authors · 2018-11-15
> **Response to Reviewer 3**
>
> We thank the reviewer for a careful reading of our paper and the positive comments about our work.
>
> Q: “Some parts could be improved. The results of Fig. 2B to investigate the role of latent variables extracted from the trained autoencoder are not clear, the simultaneous training could be better explained. As the authors claimed that their method allows to make an unsupervised alignment neural recording, independently of the task, an experiment on another dataset could enforce this claim.”
>
> A: In the revised version of our paper we have clarified the procedure for training the AE. It is based on a loss function that includes not only the unsupervised neural reconstruction loss but also a supervised regression loss that quantifies the quality of EMG prediction (see Eq 1). This combined training resulted in low-dimensional latent variables that were then used as inputs to a muscle predictor; this predictor performed as well as a muscle predictor based directly on the high-dimensional neural activity.
> We do agree that additional experiments, both open loop (offline) and closed loop (online), with additional animals, and involving additional tasks, are required to fully validate our results; we are currently in the process of developing and running these experiments. A vetting of the computational ideas within the machine learning community is crucial before embarking into extremely time-consuming experiments.

---

### Public Comment · (anonymous) · 2018-10-08
**This work proposed a method that have been done by a Nature Method paper**

Hi:

The idea of using an encoder and decoder and a small latent spaces to design a brain machine interface has already been done by this paper:

Inferring single-trial neural population dynamics using sequential auto-encoders
https://www.nature.com/articles/s41592-018-0109-9

Could you elaborate the difference between yours and their paper?

---

> ### Author Response · Authors · 2018-10-10
> **Our work uses a very different method; we knew of and cited the Nature Methods paper**
>
> Thank you for your comment. The paper that you mentioned in your comment (which we will refer to it as the LFADS paper) is an important work that we are very familiar with. We cite this work in our paper, referring to https://www.biorxiv.org/content/early/2017/06/20/152884, the version posted to bioRxiv in June 2017. We will update the citation to the now published version when our paper is revised.
> The LFADS paper introduces a denoising auto-encoder to extract low-dimensional latent variables from neural recordings. These latent variables are then used as inputs to a predictor of movement related variables. This aspect of our work is indeed similar to theirs. However, our goal is not to extract a latent space from the neural, a project that several BMI groups have already contributed to. Our goal is to obtain a statistically stable latent representation, one that can provide stable inputs to a fixed predictor of movement related variables.
> The need for the stabilization of the latent space arises because of continuous changes in the recording device. To address this issue, we introduced an adversarial domain adaptation technique that matches the probability distribution of the residuals of the reconstructed neural recordings across days, as a proxy to matching the probability distribution of the latent variables. To our knowledge, this is the first implementation of an adversarial domain adaptation method to successfully align latent variables across days and achieve stable predictions of movement related variables. The LFADS paper does not propose a method to compensate for the daily changes in the neural recordings; they deal with this instability by continuing to train the interface over as long as five months. As we write in our paper when describing Related Work: “Pandarinath et al. (2017) extract a single latent space from concatenating neural recordings over five months, and show that a predictor of movement kinematics based on these latent signals is reliable across all the recorded sessions.” As we discuss in our paper, this is not a viable solution in practical applications, because it requires the user to continuously adapt to a changing interface.
>
> To summarize, there is no overlap in the design of the interface between our manuscript and the LFADS paper beyond the fact that in both papers an auto-encoder is used to reduce the dimensionality of the recorded neural signals. The idea of extracting a latent space through dimensionality reduction is not new. In recent years it has become well established that there is a high degree of correlation across neural signals recorded from the primary motor cortex (M1); the practice of extracting a low-dimensional latent space from neural recordings has thus become quite common among many BMI groups. The LFADS paper is a recent and important publication on this topic, joining a relatively large number of preceding studies, such as Yu et al., 2009; Shenoy et al., 2013; Sadtler et al., 2014; Gallego et al., 2017a (see the full citations in our manuscript).
> Although both the LFADS paper and our paper achieve dimensionality reduction through an auto-encoder, the architectures of the two networks are very different. LFADS is a sequential, variational auto-encoder with two RNNs, based on the assumption that spikes are samples from a Poisson process. In contrast, we have implemented a simple feed-forward auto-encoder architecture. We thus emphasize the statistics of the latent variables as opposed to their dynamics.
> Yet another difference between the interface presented here and LFADS is that we simultaneously train the neural auto-encoder and the network that predicts movement related variables from latent variables. LFADS uses a sequential approach of first extracting the latent space followed by training a movement predictor. We provide evidence in our paper that the supervision of the dimensionality reduction step through the integration of relevant movement information leads to a latent representation that better captures neural variability related to movement intent and therefore significantly improves the performance of the interface.

---

### Author Response · Authors · 2018-11-19
**Revision Available**

Dear reviewers,

We have submitted a revised version of our paper. We have done our best to address all your comments, and hope you will find the changes to be positive. We would be glad to address any additional or remaining concerns.  Thank you for your feedback and comments; the revisions you suggested have strengthened the paper.

---

### Meta-Review · Area_Chair1 · 2018-12-12
**nice**

**Confidence:** 4
**Recommendation:** Accept (Poster)

**Metareview:**

BMIs need per-patient and per-session calibration, and this paper seeks to amend that.  Using VAEs and RNNs, it relates sEEG to sEMG, in principle a ten-year old approach, but do so using a novel adversarial approach that seems to work.

The reviewers agree the approach is nice, the statements in the paper are too strong, but publication is recommended.  Clinical evaluation is an important next step.